# Impact of Social Media on Adolescence: Mapping Emerging Needs to Build Resilient Skills

Carolina Falcón-Linares [1,*] , Sara González-Yubero [1] , Marta Mauri-Medrano [1]
and María Jesús Cardoso-Moreno [2]

1   Department of Educational Sciences, University of Zaragoza, 50009 Zaragoza, Spain;
    sara.gonzalez@unizar.es (S.G.-Y.); mmauri@unizar.es (M.M.-M.)
2   Department of Psychology and Sociology, University of Zaragoza, 50009 Zaragoza, Spain; mcarmor@unizar.es
*   Correspondence: cfalcon@unizar.es

**Abstract:** It is important to study the impact of social media on mental health and well-being, as most young people use social media. Research has provided evidence of the link between social media and mental health, identifying vulnerability variables, risk factors, comorbidity, and predictors of deterioration or improvement. However, there is still very little qualitative insight into young people's experiences and perceptions of social media and its impact on their subjective well-being. This study consists of a systematic review of the literature and a narrative synthesis of scientific articles published between 2013 and 2023 and indexed in the most important scientific databases in our field of knowledge. The SALSA protocol for systematic reviews of scientific literature was followed. We worked on a final sample of 25 articles, all of which were qualitative in methodology. From the content analysis, we extracted five thematic categories that describe and explore in depth the complex impact of social networks on adolescents' well-being. The interactions between positive and negative effects, as well as the links with protective or vulnerability factors, are presented with the aim of constructing as complete a knowledge framework as possible. The paper concludes with useful implications for educational interventions.

**Keywords:** adolescence; social media; resilient skills; resilient intelligence; mental health; qualitative research; systematic review

## 1. Introduction

Social media are digital platforms that allow people to interact by sharing, commenting, and responding to content [1]. As the majority of adolescents use social media, it is important to investigate its impact on their mental health [2]. It is thought that the motivation for using the network and the time spent online could be useful for characterization. It is also worth noting that most patients present with comorbidity between multiple disorders, which further complicates their description and understanding [3].

The concept of 'psychological well-being' reflects the extent to which a person can live a meaningful life in accordance with their deepest values [4]. Mental health has been defined by the WHO [5] as a state of well-being that enables a person to fulfil his or her potential, interact positively with society, and contribute to society. Research in this area is traditionally quantitative, providing evidence of the relationship between social networks and mental health, with limited insight into the experiences and beliefs of adolescents about their use and impact of social networks.

Over the past decade, depression has become more common among adolescents and young adults, according to international reports [6]. Over the same period, the use of social media has also increased. It is neither possible nor relevant to claim that social networking causes depression, anxiety, or addiction. However, there is considerable recent scientific evidence to suggest that social media use can be harmful to adolescents and even younger people [7]. In addition, peer-to-peer online relationships are less satisfying than face-to-face

relationships [8]. Studies show that teenagers who spend more time on social networking sites also feel lonelier. Children who already feel lonely may also use social networks more. However, it could also be that using social media makes social relationships worse [9,10].

An alternative theory is that social media is bad for the self-esteem of young people. Exposure to lots of perfect photos online can make teenagers (particularly women) feel bad about their appearance [11]. Depression and further isolation can result from feeling bad about oneself. Key vulnerability factors described in research include earlier age of connection to social networks, having emotional lability, low self-esteem, unstable personality, embarrassment, insecurity, family deficiencies, and poor guidance [12]. Given the high prevalence of comorbidity, mental health professionals question whether addiction is a disorder in itself or a manifestation of an existing primary disorder [13]. For instance, comorbidities have emerged between online dependence and depression, ADHD, social anxiety disorder, and challenging behaviour. In addition, internet addiction in children and adolescents is linked to clinical increases in obsessions and compulsions, deficits in social skills, feelings of being alone, antisocial behaviour, isolation, and, most importantly, the depth of depression [14–16].

Social media also reduces the amount of time that young people have to engage in leisure and other things that help them to be happy, such as exercise and hobbies. They also take their attention away from important tasks such as school, sports, or art. Juggling these responsibilities in order to have time for everything can lead to increased stress and worries about poor time management, given the significant proportion of time that goes into networking [3]. Studies also suggest that late night use of social media interferes with sleep, which for many young people is insufficient and less restful [17].

As explained above, the predominant research in this area to date has used quantitative methodology, establishing the relationship existing between digital environments and psychological well-being, identifying variables of vulnerability, risk factors, comorbidity, and predicting deterioration or improvement. It has mainly been approached from a health care perspective and published in clinical psychology media. However, there is still a very limited amount of qualitative insight into the experiences of young people with social networks and their effects on the subjective well-being of young people. We have not found an in-depth body of knowledge that addresses their educational needs and provides evidence for decision making in our field. In order to contribute to improving the quality of education, the research presented here aims to demonstrate the complex impact of social networking on adolescents' emotional well-being from their own perspective.

## 2. Method

In this study we followed a systematic literature review methodology using the SALSA protocol for systematic reviews of qualitative studies (Table 1). This is a method for selecting, filtering, synthesizing, and appraising evidence from the qualitative scientific literature [18,19]. It aims to find common views, repeated patterns of behaviour, or concepts across or between qualitative reports [20]. The search starts by looking for articles from the last ten years that have the following key descriptors: 'social networking', 'mental disorders', 'well-being', 'adolescents', 'internet addiction', 'connectedness', and 'bullying'. To make these searches more specific, two Boolean filters (AND/OR) are used to include several similar terms: (1) perceptions/attitudes/views/opinions, (2) adolescents/youth/teenagers, (3) social media/networks/Instagram/Facebook/Snapchat/Tiktok/Twitter, (4) mental health/well-being/anxiety/depression/self-esteem. The following databases were searched: APA PsychInfo, Web of Science, Scopus, and PubMed.

The initial search results are reviewed to remove papers that do not use qualitative methods, or whose internet use is broader than social network interaction. Furthermore, because of the multi-faceted nature of factors related to mental health during COVID-19, those papers that examined social network use during this period were also excluded. Out of 77 documents found, 57 are excluded due to the above criteria and 20 articles are selected for content analysis. After this first register of valid articles, five additional articles are

identified from searches in the reference and citation lists of the twenty selected articles. Thus, a sample of 25 research articles is worked on. The vast majority of the 25 articles are contextualized in Western cultures. Only three of them were studied with samples from Eastern cultures. However, there are common threads in the articles, so that the selection of articles analyzed responds to the interests of an international audience.

**Table 1.** Diagram of the SALSA systematic reviews protocol for obtaining the sample of articles (self-development).

| Search Research Articles from 2013 to 2023 Keywords: 'Social Networking', 'Mental Disorders', 'Well-Being', 'Adolescents', 'Internet Addiction', 'Connectedness' and 'Bullying' Bolean Operators (AND/OR): (1) 'Perceptions/Attitudes/Views/Opinions', (2) 'Adolescents/Youth/Teenagers', (3) 'Social Media/Networks/Instagram/Facebook/Snapchat/Tiktok/Twitter' & (4) 'Mental Health/Well-Being/Anxiety/Depression/Self-Esteem' | | | |
|---|---|---|---|
| **APA PsychInfo Documents Found: n = 53** | **Web of Science Documents Found: n = 65** | **Scopus Documents Found: n = 59** | **PubMed Documents Found: n = 17** |
| Elimination of overlaps = 117 | | | |
| AppraisaL N = 77 Exclusions for: quantitative studies &/OR internet usage not in social media &/OR COVID-19 framework = 57 → N = 20 Added from the references of the 20 selected articles = 5 | | | |
| Synthesis and Analysis N = 25 | | | |

Qualitative findings from each article are coded openly (without pre-determined categories or hypotheses) and recurring themes are identified from content comparison. Narrative categories resulting from the analysis are extracted and synthesized using a thematic synthesis methodology [21].

## 3. Results

As a product of the whole analytical and synthesis process, five thematic categories are identified (Table 2). These five categories are thought to describe the processes that influence mental disorders and well-being by interacting in social networks over the time frame of this theoretical review. The contents linked to each category are described below with the most relevant bibliographic citations.

**Table 2.** Name and overview of emerging categories (self-development).

| Thematic Categories | Summary Definition |
|---|---|
| Self-expression and social validation | Adolescents optimize the representation of themselves in social media and feedback from peers usually leads to profile modification. Expectations of validation and impulsive behaviour by publishing are very relevant variables in the study of psychological symptoms. |
| Overestimation of physical appearance and pursuit of perfect look | The content in social media is predominantly image-based, so the look is very important, affecting girls significantly more. The 'ideal figure' can be especially damaging for people suffering from pre-existing eating-related pathologies. Anxiety regarding dependence on validation is also among the most studied factors. |
| The stress of constant networking to remain online and its consequences | Social media activity dominates young people's time management and becomes an essential element of everyday life. The idea of disconnecting induces fear, and the risk of social exclusion fuels compulsive use. There has been a reduction in the amount of satisfying time spent with relatives and peers. |

**Table 2.** *Cont.*

| Thematic Categories | Summary Definition |
|---|---|
| Getting involved and supporting each other | Networks can make a beneficial contribution to well-being by helping and getting involved. Online friendships correlate with psychological resilience variables against anxiety, such as perceived support. Also, there are initiatives that aim to promote positive mental health and overcoming fears about seeking professional help. |
| Facing cyberbullying or models inciting self-harm | Cyberbullying, abusive language, and shaming cause a lot of panic attacks and chronic anxiety in this population. One of the riskiest aspects is the anonymity in which bullies can remain. Another aspect is that episodes are sustained over time. Moreover, exposure to harmful content is frequently described, including publications of self-harm procedures. |

### 3.1. Self-Expression and Social Validation

One of the most repeated themes is that adolescents report the potential of social media to optimize the representation of oneself. Young people are building their identities so that they represent the best version of themselves [22]. Although self-expression is potentially perceived as an exercise in freedom, many admit to the determining influence of others' opinions. Feedback from peers usually leads to profile modification. Although they express enthusiasm when they receive 'Likes' from other users, this pleasurable feeling is short-lived and is followed by sometimes compulsive checks on how many 'Likes' and 'Comments' they immediately receive [23,24]. Adolescents often compare their number of 'Likes' to their friends, so getting fewer can have a negative effect on their confidence and feelings. For teens with a history of fear, distress, or depression, this process increases symptoms [23,25,26].

Posting selfies is a common practice. Girls experience higher expectations of validation when posting and are concerned about, or at least consider, privacy. Boys believe that posting selfies can increase their popularity and admit their desire and need to attract "Likes". They are more impulsive in posting photos and confess to being less concerned about privacy and the appropriateness of the content [27]. Because posting selfies is often a way of validating oneself, there are unspoken codes about how to post selfies, which create an anxious hypervigilance: you have to publish as much as you are visible, which also means not posting too much, lest you be judged [22,26].

### 3.2. Overestimation of Physical Appearance and Pursuit of Perfect Look

On picture-sharing sites, in which the content is predominantly image-based, the look is very important. Flawless photos with dozens of 'Likes' are a sign of high approval ratings. In this category, there is one factor common to all studies with reference to this topic: gender differences. For girls, 'Likes' are an endorsement of conformity to certain physical standards, which have their roots in the way perfection is portrayed in social networks [28]. Image-based social networks are perceived as being most harmful to their sense of self-worth with a clear gender gap, affecting girls significantly more. In addition, the practice of retouching images is common and fuels expectations of physical idealization. Young people of both genders are aware that celebrity photos are retouched, and yet they continue to conform one's real appearance with those that seem more 'ideal'. It explicitly lowers their confidence, creates unworthiness, and has a negative influence on their perception of their own person [11,29,30]. There are so many of these pictures on the internet that it is very hard not to find them. In addition, it is considered that pictures representing the 'ideal figure' can be especially damaging for people suffering from pre-existing eating-related pathologies, so this practice leads to worsening of the symptom picture [24], as is the case with anxiety regarding the dependence on profile validation.

Although girls feel scrutinized more often, boys also experience criticism and recognize its influence on self-esteem. They report that publishing photos of themselves in genderless or unisex clothes has triggered comments that have questioned their sexuality and have therefore been experienced as negative and limiting to their sense of freedom [28].

### 3.3. The Stress of Constant Networking to Remain Online and Its Consequences

Young people often describe the way social media activity dominates their time management and becomes an essential element of everyday life. They feel the need for constant interaction, e.g., by taking part in 'threads' of posts, and believing it is necessary to post something daily to maintain a high number of interactions [30]. The idea of disconnecting induces fear, because they fear not being aware of what is happening on networks and feel at risk of social exclusion [22,31,32]. This fuels compulsive use. Constantly picking up their phones and checking multiple social networking apps, one after the other, just for the sake of feeling that they are still connected, is described as a reflexive action [30,31]. The simple gesture of opening the apps again and again induces a sense of calm, with this response resembling that of compulsive checking in some typologies of obsessive-compulsive disorder [33].

On the other hand, there is a widespread perception that there has been a reduction in the amount of satisfying time spent with relatives and peers, causing a sense of disconnectedness with authentic and loved people [34]. Adolescents experience communication problems within the household and realize that personal contact with parents has become impoverished [22]. In addition, to a lesser extent, they report symptoms caused by their use of electronic devices, such as migraines, disturbed sleep, and visual impairment [35].

Thomas et al. [32] investigated the experimental situation of disconnecting from social networks for a whole day. Some respondents were concerned and uncomfortable with the lack of information about their disconnection. Anyone who had been trained for a few hours to disconnect, found it much less anxious, produced higher performance in their activities, and provided more ideas about what to do and who to interact with while disconnected. A common feature in almost all textual accounts of research on this topic is that adolescents talk about their difficulties with online disconnection as if they were talking about other people, not themselves, suggesting they normalize their behaviors, unlike the so-called 'mobile/network addicts'. As a result, they may be avoiding face-to-face contact to the extent that they are utilizing the Internet for socializing and to self-evaluate in a biased way.

### 3.4. Getting Involved and Supporting Each Other

Young people describe the way in which networks can make a beneficial contribution to well-being, by helping and getting involved. Relationships blossom and grow through tweets, texts, and posts [36]. Several articles reiterate the value of having a digital footprint in creating opportunities for community participation [8,24,32,37]. The argument is that there is a simplicity to making contacts remotely, without the stress of face-to-face meetings. Online friendships correlate with psychological resilience variables against anxiety, such as perceived support [8]. Adolescents quantitatively value their 'friends' or 'followers', and those with higher numbers express that this increases their self-esteem, acting as a variable that validates or demonstrates their popularity. But while having a larger network gives them comfort, they do not always feel that they matter to their 'friends' in networks. Most say that non-networked or 'real life' friends have greater value [8,24].

In this thematic category, as a positive counterpoint to the others, interaction initiatives that aim to promote positive mental health are noteworthy. For example, participants in O'Reilly et al.'s study [30] describe acting on social media in terms of "challenges" to improve their well-being, such as publishing pictures every day of personal sources of inspiration. Next, people invite their contacts to join in by 'tagging' several of them to 'share' the positive message with their peers. Using the network also provides an opportunity to talk about psychological disorders and well-being. They are a good channel for disseminating celebrity recovery stories, which reduces isolation for individuals facing similar experiences and enables those diagnosed to form supporting communities [30,38]. In addition, several studies have demonstrated the benefits of using collaborative chat tools, facilitated by health experts, for people with emotional and behavioral difficulties [39,40].

Users value the huge support network and the secure environment in which they can talk about anything.

Ultimately, it is essential to highlight that these forums provide support in overcoming fears about seeking professional help, providing examples where this first step has helped many people in their recovery [40]. Many young people emphasize the importance of improved counselling in education in order to get the most out of using networks; they need to find reliable information and appropriate groups to join [30,41].

*3.5. Facing Cyberbullying or Models Inciting Self-Harm*

Despite the benefits of social media, young people report feeling more vulnerable to bullying. In addition, digital content posted to cause harm is estimated to have a very serious effect on emotional equilibrium. One of the riskiest aspects is the anonymity in which bullies can remain, posting offensive messages with impunity as their real identities remain unknown [18,26]. Cyberbullying, abusive language, and shaming cause a lot of panic attacks and chronic anxiety in this population [23]; however, there is a widespread belief that this harmful content is to be accepted, ignored, or tolerated [30,42].

Adolescents who have experienced cyberbullying report a harmful effect on psychological functioning, reporting symptoms such as loneliness and uncertainty [30,43]. While many are aware of the injustice of such content against their dignity, they do not express confidence about the sole responsibility of perpetrators, and many do not know where to turn [43]. When episodes are sustained over time, and certain personal variables are coupled with increased vulnerability, including female gender, the consequences are incredibly damaging, sometimes leading to suicidal or self-harming thoughts and behavior [37]. Some bullies admit to the expressed intention of lowering their victims' self-esteem; these are behaviors that are usually underpinned by jealousy or feelings of threat, inferiority, or dissatisfaction with their own lives [28]. At another level of severity, though not insignificant, subtle harms caused through exclusion are also described, including failure to invite individuals to events, respond to messages, or validate content, with the express purpose of doing harm.

There are also common worries about confidentiality. Despite the fact that you only make your profile public if you want to, teenagers tend to post publicly because they are afraid of being excluded and, once their images have been shared, they are afraid that others will share their photos without their permission. This lack of control induces anxiety, which is made worse by the perpetuation of content after uploading it to the cloud [42].

Finally, exposure to harmful content is frequently described, including publications of self-harm, self-harm procedures, and how to avoid being easily identified by family members or teachers. Strategies are proposed and advice is offered for these practices, which, considering the immaturity of adolescents, are extremely dangerous [7,24,26].

## 4. Discussion

This article describes the complex impact of networks on the psychological well-being of young people, considering their beliefs, perceptions, feelings, etc. The motivation of the study is to build a current and in-depth knowledge framework on the topic, which serves as a scientific foundation for educational intervention. It is essential to address the effect that the virtualization of social relations is producing in this evolutionary stage from the perspective of its protagonists, in order to be able to adopt decisions that involve both their educators and educational policy and interdisciplinary professional coordination. The methodology used is justified by the need to delve deeper into the relationship that adolescents establish with social networks, from an emergent and non-deterministic perspective.

This discussion is approached from a dual perspective that does not seek to negatively judge the use of social networks, but rather to provide evidence of how their use should be oriented in order to take advantage of their positive potential and reduce their risks. Ultimately, the aim is to achieve responsible and sustainable use. The fact that adolescents

talk about mental health in a natural way, to which these channels contribute, is a good start to break down stereotypes and the stigma that many people have mental health problems. However, the qualitative results synthesized in this study support and strengthen the well-known negative effects that new styles of social relations have on health, which primarily affects the investigated population [2].

From the narrative analysis by category, a positive evaluation of the networks emerged, highlighting that they produce and facilitate peer relations and supporting relationships. Digital contacts are, in fact, a mode of social connection that increases feelings of integration and social reinforcement. In terms of their good uses and practices, these channels are seen as spaces for learning from each other, sharing ideas and resources, and dealing with difficulties. In this context, it was striking to find reports of their contribution to health and well-being through participation in groups that share the same mental disorders. In the same way, debate groups, moderated by psychologists, guide discourse on the most common issues, reduce feelings of loneliness, and encourage people to seek professional help from the health system.

However, it has been widely and deeply evidenced that using social media can have a damaging effect on well-being and mental health. Matalí-Costa's findings have been reinforced and extended [3], with other publications detailing damage to self-esteem, and those detailing how emotional harm is more damaging to people with vulnerable personal characteristics [7,33], even provoking or facilitating self-harming and suicidal behaviour [12]. In parallel, the compulsive use of networks negatively impacts how they relate to each other in real life [9,10] or causes insomnia and fears [17]. Another well-documented finding is the negative use of social networks for cyberbullying. Although there is a close link between cyber and in-person forms of bullying, studies have shown that anonymity, the power to spread, and the permanence of humiliating content, generate more extreme levels of victimization and chronicity of anxious-depressive impact on victims.

It is important to note that the resulting thematic categories significantly interact with one another, reflecting the multi-faceted nature of the variables involved. Several of the studies analyzed show opposing descriptions and arguments, with pros and cons of network use, even for the same individuals. For example, there are descriptions of experiences of the positive use of platforms that have generated anxiety about staying connected, fueling addiction. This problem of anxiety about disconnection is one of the most widespread and coinciding problems in the research analyzed.

An important challenge is to move away from viewing adolescents as a homogenous population. This study includes samples of adolescents aged 13–17 years, and it highlights the great diversity within this age group, which magnifies the complexity of digital impact over various psychosocial factors. Everyone has different levels of 'digital skills'. This is crucial for safety and security when using digital tools, such as networks [44]. Livingstone [45] has emphasized that negative impact depends on the intricate interactions between personal traits of vulnerability or resilience and being in protective or stressful environmental contexts, leading to important differences among people subjected to the same networking events or experience. Specifically, self-confidence and competent parenting styles in dealing with social media are considered resilience drivers [46], whereas having a history of any kind of emotional distress makes people more susceptible to social network content and use. In addition, individuals with greater digital resilience may be better able to recognize and reduce the impact of exposure to digital threats. What is important is that people who are prone to affective imbalance outside the digital environment also have a higher likelihood of being at risk online [47]. Therefore, in order to better understand the connection between 'real life' and the circumstances/experiences that worsen mental health in the digital environment, more research is needed.

Drawing on social comparison theory [48], there is a tendency to evaluate one's own ideas and skills by comparing oneself with others. Curiously, this pattern is more common in adolescence than in childhood or adulthood [49,50]. How social networking affects mental health seems to vary from one individual to another depending on the referents

used for self-evaluation and the motivation behind their participation in social networks. For example, regarding motivation and type of network use, Appel et al. [9] find that the use of networks from the role of observers who are not very active, predicts attitudes of jealousy and negative social self-evaluation, which is in turn related to a worsened self-concept and depressive state. Moreover, girls have been found to experience higher expectations of validation when posting their photos and content. They are also more concerned about privacy, or at least consider it. Boys believe that posting images of themselves can increase their popularity and admit their desire for validation, sometimes even the need, to attract 'Likes' as validation of their content. However, boys are more impulsive and not as concerned about privacy or the permanence of what is on networks [22,26].

As has long been well known, the most important period for personality development and self-fashioning is adolescence [51], and much of this is currently driven by social media relationships. Given their poor ability to self-regulate and susceptibility to group stigma, teenagers are most vulnerable towards exposure to the harmful consequences of using digital networks and are therefore more likely to become mentally disordered [36]. Placing the evidence in the broader context of how individuals develop cognitively and socially during adolescence helps to provide a window into understanding the processes that determine how networks affect psychological well-being during this period. During the teenage years, the brain undergoes profound changes. The prefrontal area, which is responsible for the resolution of higher cognitive demands, has yet to finish developing in this period, leading adolescents to make decisions through a fast-processing pathway with a strong emotional component [2]. In addition, the nucleus accumbent, which regulates dopamine-modulated reward, is overactive [52]. As a result, young people are highly susceptible to decisions made in the spur of the moment, which poses a problem in the digital world, in which sharing is instantaneous, at the touch of a button. In terms of evolutionary mechanisms, during adolescence, synaptic pruning selects neural networks, which simplifies and reinforces certain patterns of reasoning that will persist throughout life [53], which can lead to an impoverished conception of one's self-image. And, more seriously, this can affect self-injurious and suicidal thought patterns [37].

The results of this theoretical analysis also lead to a reflection on the veracity of Erikson's theory of social development, according to which, at the developmental stage in focus, the greatest stress is due to dealing with two processes intrinsic to the growth of identity: feeling that one fits in with oneself and with the group [51]. In this sense, networks provide a forum to engage in this battle for self-determination, with the context and channels changing at a high speed. For his part, Lewin identifies this stage as a tense life phase characterized by intrapersonal forces resulting from being exposed for the first time to a social environment without parental supervision [54]. This environment, since the emergence of social networks, is duplicated or divided, with a real and a virtual scenario, which could lead to living these dynamics in parallel, constructing two identities, the real and the virtual one. This approach has good and bad connotations. Regarding the positive ones, networked interaction initiatives that aim to promote positive mental health are noteworthy. They are a good channel for disseminating celebrity recovery stories, which reduces isolation for individuals facing similar experiences, and enables the perception of peer support [38]. Users value the fact of feeling accompanied by other people diagnosed with the same disorders. Importantly, these sites encourage people to seek specialist assistance and overcome the prejudice surrounding mental illnesses, providing examples where this first step has helped many people in their recovery [40]. Also noteworthy is their potential for learning, considering that the usefulness and aesthetic appeal of social networks significantly influence learners' knowledge sharing and seeking behavior [55]. The risk is not being sufficiently educated to discriminate appropriate sources of knowledge.

However, as a counterpoint, many adolescents have reported experiencing verbal abuse and being subjected to humiliating comments that lead to crises and chronic anxiety. Furthermore, there is a widespread belief that this harmful content is to be accepted, ignored, or tolerated [30,42]. At another level of severity, though not insignificant, subtle

harms caused through exclusion, including failure to invite individuals to events, respond to messages, or validate content, with the express purpose of doing harm, are also described.

This qualitative scientific literature review has underlined the multifaceted implications of social media on young people's well-being. The review has allowed for an advanced knowledge of the reasons underlying the beneficial and detrimental impacts of digital life on emotional well-being. The answer lies in education. A useful framework of knowledge for teachers and school counsellors can be extracted from the discussion. The challenge is to enhance the good that these channels bring and to minimize or mitigate their negative and dangerous effects. An example of this can be found in an educational program developed for secondary education, which uses social networks as a channel to stimulate actions of gratitude with the aim of improving the well-being of students [56]. Educational action should focus on the use of social networks as an additional educational tool. Social networks are used by young people in vital contexts and occupy a large part of their time. There is an urgent need to support with scientific evidence some good practices that are beginning to be implemented in educational institutions by teachers and counsellors. The strongest theoretical reference leads us to consider positive psychology as a basis for designing interventions.

**Author Contributions:** Conceptualization, C.F.-L. and M.J.C.-M.; methodology, C.F.-L.; formal analysis, C.F.-L., S.G.-Y. and M.M.-M.; data curation, C.F.-L., S.G.-Y. and M.M.-M.; writing-original draft preparation, C.F.-L.; writing-review and editing, S.G.-Y., M.M.-M. and M.J.C.-M. All authors have read and agreed to the published version of the manuscript.

**Funding:** This research was funded by Erasmus Plus Project GRIT: Growth mind-set through Resilient Intelligent Technologies (KA220-HED-F3A026B8).

**Conflicts of Interest:** The authors declare no conflict of interest.

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
