# Peer review of "Impact of Social Media on Adolescence: Mapping Emerging Needs to Build Resilient Skills"

_societies, doi:10.3390/soc13110238_

Round 1

Reviewer 1 Report

Comments and Suggestions for Authors

First of all, I would like to congratulate the authors for their interesting work. I would recommend a summary table with the five categories proposed as a result of their research.  I would enhance their work and make it easier for the reader to understand.

I find the example of the proposed educational program for adolescents in secondary education on the use and abuse of social networks very interesting.

Author Response

Dear Reviewer,

Thank you very much for your review and your recommendation. Please find the table you have indicated and the reference within the text.

Kind regards,

The authors

Reviewer 2 Report

Comments and Suggestions for Authors

This is a very good overview of the literature on an important topic, and that is valuable.  However, I wish the article added something new.  I would like to see a mention of how culture does or does not affect the impact of social media on young people's health and well-being.  Did the 25 articles referenced relate primarily to Western audiences, Eastern audiences, or international audiences? The last paragraph needs development.  Just saying that education is the answer is not enough.  Provide a bit more direction.  Finally, on page 3, line 119, use "fewer" rather than "less."  "Fewer" refers to how many (as in "likes"); "less" refers to how much.  Thanks for your good work.

Author Response

Dear Reviewer,

Thank you very much for your review and your recommendations.

Content added to the manuscript in response to the following requests:

  • I would like to see a mention of how culture does or does not affect the impact of social media on young people's health and well-being.  Did the 25 articles referenced relate primarily to Western audiences, Eastern audiences, or international audiences?

“The vast majority of the 25 articles are contextualised in Western cultures. Only 3 of them were studied with samples from Eastern cultures. However, there are common threads in the articles, so that the selection of articles analysed responds to the interests of an international audience.”

  • The last paragraph needs development.  Just saying that education is the answer is not enough.  Provide a bit more direction. 

“Educational action should focus on the use of social networks as an additional educational tool. They are in the vital context of young people and occupy a large part of their time. There is an urgent need to support with scientific evidence some good practices that are beginning to be implemented in educational institutions by teachers and counsellors. The strongest theoretical reference leads us to consider positive psychology as a basis for designing interventions.”

  • Finally, on page 3, line 119, use "fewer" rather than "less."  "Fewer" refers to how many (as in "likes"); "less" refers to how much.  Thanks for your good work.

          It has been corrected, thank you very much.

Kind regards,

The authors